

# The diagnostic value of serum insulin-like growth factor binding protein 7 in gastric cancer

Can-Tong Liu[1,2,3,*], Fang-Cai Wu[2,4,*], Yi-Xuan Zhuang[5], Xin-Yi Huang[6], Xin-Hao Li[1], Qi-Qi Qu[1], Yu-Hui Peng[1,2,3], Yi-Wei Xu[1,2,3], Shu-Lin Chen[7] and Xu-Chun Huang[1,2,3]

[1] Department of Clinical Laboratory Medicine, the Cancer Hospital of Shantou University Medical College, Shantou, Guangdong, China
[2] Esophageal Cancer Prevention and Control Research Center, the Cancer Hospital of Shantou University Medical College, Shantou, Guangdong, China
[3] Guangdong Esophageal Cancer Research Institute, Guangzhou, Guangdong, China
[4] Department of Radiation Oncology, the Cancer Hospital of Shantou University Medical College, Shantou, Guangdong, China
[5] Department of Pathology, the Cancer Hospital of Shantou University Medical College, Shantou, Guangdong, China
[6] Department of Gastrointestinal Endoscopy, the First Affiliated Hospital of Shantou University Medical College, Shantou, Guangdong, China
[7] Department of Clinical Laboratory Medicine, Sun Yat-Sen University Cancer Center, Guangzhou, Guangdong, China
* These authors contributed equally to this work.

Corresponding authors
Shu-Lin Chen,
chenshl@sysucc.org.cn
Xu-Chun Huang,
313623874@qq.com

## ABSTRACT

**Backgrounds:** Early detection might help in reducing the burden and promoting the survival rate of gastric cancers. Herein, we tried to explore the diagnostic value of insulin-like growth factor binding protein 7 (IGFBP7) in gastric cancers.

**Methods:** In this study, we first analyzed the expression levels and prognostic value of IGFBP7 mRNA in gastric cancers from The Cancer Genome Atlas (TCGA) database. Then, we recruited 169 gastric cancer patients and 100 normal controls as training cohort, and 55 gastric cancer patients and 55 normal controls as independent validation cohort. Enzyme-linked immunosorbent assay was applied to test the serum levels of IGFBP7. The receiver operating characteristic curve (ROC) and the area under the curve (AUC) were applied to evaluation the diagnostic value.

**Results:** TCGA showed that IGFBP7 mRNA was dysregulated and associated with prognosis in gastric cancer patients. Then, we examined the expression of serum IGFBP7 and found that serum IGFBP7 expressed lower in gastric cancer patients than normal controls both in training and independent validation cohorts ($p < 0.0001$). In training cohort, with the cutoff value of 1.515 ng/ml, the AUC for distinguishing gastric cancer patients was 0.774 (95% CI [0.713–0.836]) with sensitivity of 36.7% (95% CI [29.5–44.5]) and specificity of 90.0% (95% CI [82.0–94.8]). As for early-stage EJA, the AUC was 0.773 (95% CI [0.701–0.845]) with the sensitivity of 33.3% (95% CI [14.4–58.8]). In independent validation cohort, with the same cutoff value, the AUC reached to 0.758 (95% CI [0.664–0.852]). Similarly, for early-stage gastric cancer diagnosis in the independent validation cohort, the AUC value was 0.778 (95% CI [0.673–0.882]).

**Conclusions:** This study indicated that serum IGFBP7 might act as a potential early diagnostic marker for gastric cancers.

# INTRODUCTION

Gastric cancer is one of the most prevalent malignancies over the world, accounting for the sixth incidence and fourth mortality of all cancers according to the 2020 global cancer statistics (*Sung et al., 2021*). Although the incidence rate of gastric cancer has decreased in the past few decades, the five-year survival rate is still very low. It is estimated that the five-year survival rate of patients with advanced gastric cancer is 10–20% (*Rawla & Barsouk, 2019*). If gastric cancers could be early detected, the five-year survival rate of these patients might increase to over 90% (*Luo & Li, 2019*). In the countries with high human development index, the incidence and mortality of gastric cancer were 2- or 3-times higher than those of countries with low human development index, which might benefited from their positive methods on early gastric cancer detection (*Yao et al., 2020*). If the incidence and mortality rates are steady, it is estimated that the burden of gastric cancer would increase to 1.77 million new cases and 1.27 million deaths in 2040 (*Morgan et al., 2022*). Therefore, it emphasizes the importance of early detection and diagnosis in gastric cancer. In recent clinical practice, the diagnosis of gastric cancer mostly depends on endoscopic tissue excision and pathological biopsy, which are the invasive methods (*Committee et al., 2012*). Some serum tumor markers, like carcinoembryonic antigen, cancer antigen 19-9 and cancer antigen 72-4, has been used for gastric cancer diagnosis, prognosis, treatment monitoring and recurrence detection (*Kotzev & Draganov, 2018*). However, it was reported that their early diagnosis values were not adequate (*Guo et al., 2020*). Thus, it is urgently needed to find some novel tumor markers for early detection and diagnosis of gastric cancers.

The insulin-like growth factor binding protein (IGFBP) super family contains six IGFBPs and 10 IGFBP-related proteins (IGFBP-rPs) (*Bailes & Soloviev, 2021*). Among them, the ten IGFBP-rPs include IGFBP7, cellular communication network factor 2 (CCN2), CCN3, CCN1, high temperature requirement factor A1 (HTRA1), endothelial cell-specific molecule 1 (ESM1), CCN5, CCN4, CCN6 and Kazal-type serine protease inhibitor domain containing protein 1 (KAZALD). Compared with IGFBPs, IGFBP-rPs show the low affinity to bind insulin-like growth factor (IGF), which might lead to the undefined function (*Rodgers, Roalson & Thompson, 2008*). Despite all that, IGFBP-rPs were also found to be involved in some biological and pathobiological functions. IGFBP7, also known as IGFBP-rP1, has been reported to be associated with several diseases, including cancer and acute kidney injury (*Albert et al., 2021*; *Jin et al., 2020*). As one of the first gap cell cycle arrest biomarkers, urinary IGFBP7, multiplied with tissue inhibitor metalloproteinase-2, has been approved by FDA for the acute kidney injury prediction in

patients who have acute cardiovascular or respiratory failure (*Hasson, Menon & Gist, 2022*). CCN family, containing six CCNs, have an IGFBP binding domain (*Desnoyers, 2004*). They were found to be enrolled in tumor angiogenesis, inflammatory response, fibrosis, and mitochondrial integrity (*Birkeness et al., 2022*). HTRA1 could regulate the organogenesis and pathogenesis, including cancer, through several pathways (*Oka et al., 2022*). Similarly, ESM1 was found to be involved in the Notch4 signaling pathway to regulate the tumorigenesis and progression of adrenocortical carcinoma (*Huang et al., 2021*). KAZALD, also named as IGFBP-rP10, was found as overexpression in tissue and serum of gastric cancer patients (*Shen et al., 2019*).

In this study, we first explored the expression pattern of IGFBP-rPs in gastric cancer tissues from The Cancer Genome Atlas (TCGA; https://www.cancer.gov/tcga). We found that IGFBP7 and CCN5 had significant abnormal expression between gastric cancers and adjacent normal tissues, and these two IGFBP-rPs were associated with both overall survival and disease-free survival of gastric cancer patients. Moreover, in our previous study, we have conveyed that serum IGFBP7 has the potential of early diagnosis for esophageal squamous cell carcinoma and esophagogastric junction adenocarcinoma (*Huang et al., 2019*; *Liu et al., 2020*). Therefore, in the present study, we will focus on the early diagnostic value of serum IGFBP7 in gastric cancer.

## MATERIALS AND METHODS

### TCGA database exploration

In order to find the expression pattern of IGFBP-rPs in gastric cancer, we downloaded the related mRNAs expression in gastric cancers from TCGA. Their expressions were presented with heatmaps using the R package "*pheatmap*". The differentially expressed IGFBP-rPs between gastric cancer tissues and adjacent normal tissues was defined using Mann-Whitney U test. Gene Expression Profiling Interactive Analysis (GEPIA) is a web service which integrates the TCGA and GTEx data to provide the quick analysis of cancers (*Tang et al., 2017*). We used GEPIA to apply the Kaplan–Meier survival analysis to assess the prognostic value of IGFBP-rPs. The median value of each IGFBP-rP was used to distinguish high level and low level. Log-rank test was applied to assess the significant differences.

### Study subjects

To further explore the expression of serum IGFBP7 in gastric cancers, we collected serum samples of gastric cancer patients and health volunteers from the Cancer Hospital of Shantou University Medical College and the Sun Yat-Sen University Cancer Center. Gastric cancer patients who were enrolled from May 2019 to January 2020 in Sun Yat-Sen University Cancer Center were involved into the training cohort, while those enrolled from November 2018 to April 2019 were in the independent validation cohort. Gastric cancer patients were confirmed based on endoscopy and pathological biopsy. Patients with any cancer history or cancer-related treatment history before diagnosis were excluded for analysis. Their serum samples used in this experiment were collected before surgery or any other anti-cancer therapy. Health volunteers were those with a negative upper endoscopy.
After venous sampling, blood was centrifuged at 1,250 g for 10 min, and sera were separated and stored at −80 °C until examination analysis. As for normal control groups, contemporaneous physical examination population from two hospitals were respectively divided into training and independent validation cohorts. Informed consents were signed by both patients and healthy volunteers. This study was approved by the institutional review board of the Cancer Hospital of Shantou University Medical College and the Sun Yat-Sen University Cancer Center (IRB approval number: B2022-329-01), and followed the requirements of the Declaration of Helsinki. Tumor stages were defined according to the eighth edition of the American Joint Committee on Cancer Staging Manual (*Amin et al., 2017*). In this study, stage 0+I+II was defined as early-stage gastric cancer.

**Enzyme-linked immunosorbent assay**

ELISA was carried out to examinate the serum levels of IGFBP-7. The experiment protocol was performed based on the manual from Cusabio (Catalog number: CSB-E17249h). The ELISA kit and serum samples were balanced to room temperature before the examination began. The IGFBP7 standard concentrations for standard curves construction were 10, 5, 2.5, 1.25, 0.625, 0.312, 0.156, and 0 ng/ml respectively. The serum samples were diluted by 1:3 according to our previous studies (*Huang et al., 2019*; *Liu et al., 2020*). A total of 100 µl serum sample or standard was added to each well and incubated at 37 °C for 2 h. The liquids were removed without washing the wells and 100 µl biotin-antibody were added and incubated at 37 °C for 1 h. After liquids removal and wells washing for three times, 100 µl horseradish peroxidase-avidin was added and incubated at 37 °C for another 1 h. The each well was washed for 5 times, and 90 µl 3,3′,5,5′-tetramethylbenzidine substrate for color development was added protected from light for 20 min followed with 50 µl stop solution for termination. The measurement of optical density (OD) values were executed using Microplate Reader (BioTek® Instruments, Winooski, VT, USA) at 450 nm with 570 nm reference within 5 min.

**Statistical analysis**

In this study, $p < 0.05$ (two-tailed) was defined as statistical significance. Most statistical analyses were carried out using Microsoft Excel 2019, SPSS 24.0 and GraphPad Prism 8.0. R package *pheatmap* was used to plot the expression heatmap. The OD values were changed to real concentrations according to the standard curves using SigmaPlot 10.0. Then unpaired Student's *t* test was used to evaluate the difference of serum IGFBP7 levels between gastric cancer patients and normal controls. The receiver operating characteristic (ROC) curve was performed to evaluate the diagnostic efficacy, and the area under the ROC curve (AUC) was calculated with the 95% confidence interval (CI). The cutoff value for distinguishing gastric cancer patients and normal controls was acquired by achieving the maximum sensitivity when the specificity was greater than 90% in the training cohort (*Boyle et al., 2011*). The Pearson's correlation analysis was used to evaluate the correlation between serum IGFBP7 and age. The unpaired Student's *t* test was applied to evaluate the association between serum IGFBP7 level and other clinical characteristics of gastric cancer patients.

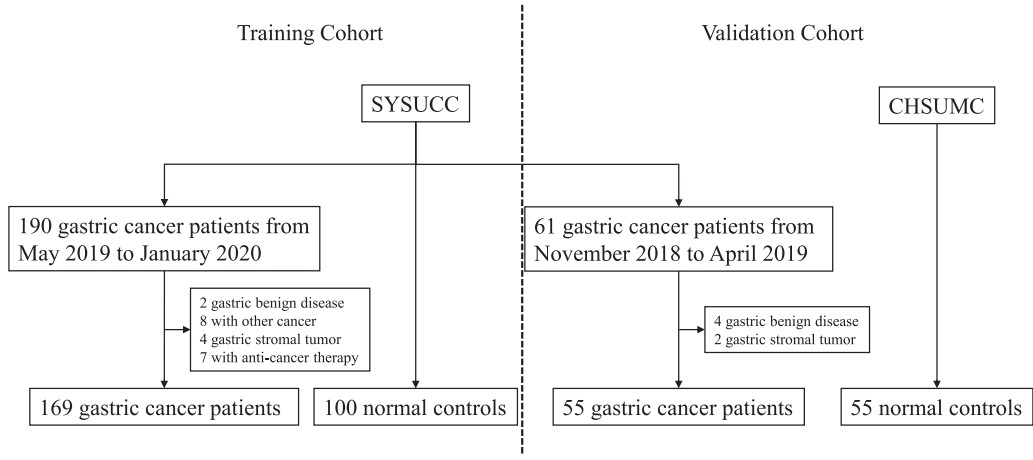

**Figure 1 Flowchart of serum IGFBP7 detection in this study.** SYSUCC: Sun Yat-Sun University Cancer Center; CHSUMC: the Cancer Hospital of Shantou University Medical College.

## RESULTS

### IGFBP-rPs expression in TCGA

Figure S1 exhibited the expression heatmap of IGFBP-rPs. Differential expression analysis showed that IGFBP7, CCN2, ESM1, CCN4 and CCN6 mRNA were over-expressed in gastric cancer tissues while CCN5 mRNA was down-expressed (all $p < 0.05$, Figs. S2A–S2J). From overall survival analysis, gastric cancer patients with high expression of IGFBP7, CCN2, CCN3, CCN1, CCN5 or CCN4 would have a poor survival time (all $p < 0.05$, Figs. S3A–S3J). However, only IGFBP7 and CCN5 was associated with disease-free survival (both $p < 0.05$, Figs. S4A–S4J). Combining both differential expression analysis and survival analysis, we found that only IGFBP7 and CCN5 were both differently-expressed and associated with prognosis. Tumor research mainly focused on those tumor related genes with overexpression, therefore, we will further explore the diagnostic value of IGFBP7 in this following study.

### Serum IGFBP7 levels

At this stage, we collected serum samples of gastric cancers patients and normal volunteers from the Cancer Hospital of Shantou University Medical College and Sun Yat-Sen University Cancer Center. As shown in Fig. 1 and Table 1, there were 169 gastric cancer patients and 100 normal volunteers in the training cohort, while there were 55 gastric cancer patients and 55 normal volunteers in the independent validation cohort. There were no significant differences between both cohorts.

Then ELISA was used to test the serum IGFBP7 protein level. ELISA test showed that in the training cohort, the mean concentration ± standard deviation of serum IGFBP7 level in gastric cancer patients was 1.580 ± 0.198 ng/ml, which was lower than in the normal control group (1.850 ± 0.328 ng/ml, $p < 0.0001$, Fig. 2A). In the independent validation cohort, the serum IGFBP7 level in gastric cancer group was 1.592 ± 0.253 ng/ml, lower than in the normal control group (2.011 ± 0.563 ng/ml, $p < 0.0001$, Fig. 2B). In early-stage

**Table 1 Clinical characteristic of gastric cancer patients and normal volunteers.**

| | Training cohort | | Independent validation cohort | | $p^*$ |
|---|---|---|---|---|---|
| | Gastric cancer ($n$ = 169) | Normal ($n$ = 100) | Gastric cancer ($n$ = 55) | Normal ($n$ = 55) | |
| Age | | | | | 0.295 |
| ≤50 | 59 | 37 | 15 | 30 | |
| >50 | 110 | 63 | 40 | 25 | |
| Gender | | | | | 0.908 |
| Male | 109 | 55 | 35 | 35 | |
| Female | 60 | 45 | 20 | 20 | |
| Invasion depth | | | | | 0.251 |
| Tis+T1+T2 | 47 | | 11 | | |
| T3+T4 | 122 | | 44 | | |
| Lymph node metastasis | | | | | 0.453 |
| N0 | 52 | | 14 | | |
| N1+N2+N3 | 117 | | 41 | | |
| Distant metastasis | | | | | 0.861 |
| M0 | 134 | | 43 | | |
| M1 | 35 | | 12 | | |
| TNM stage | | | | | 0.194 |
| Early-stage (0+I+II) | 72 | | 18 | | |
| Late-stage (III+IV) | 97 | | 37 | | |
| Lauren type | | | | | 0.930 |
| Diffuse | 69 | | 22 | | |
| Intestinal | 57 | | 20 | | |
| Mixed | 43 | | 13 | | |

**Note:**
$^*p$ was acquired from Chi-square test in gastric cancer patients between training and validation cohorts.

gastric cancer patients, the serum levels of IGFBP7 were also lower when compared with normal control groups in both cohorts (both $p < 0.0001$). There were no significant differences between early-stage gastric cancer patients and all gastric cancer patients (both $p < 0.05$).

## Diagnostic value of IGFBP7

Using the ROC curve, we acquired an AUC of 0.774 (95% CI [0.713–0.836]) for distinguishing gastric cancer patients from normal volunteers (Fig. 3A) in training cohort. When the specificity was 90.0% (95% CI [82.0–94.8]), the cutoff value was 1.515 ng/ml with the sensitivity of 36.7% (95% CI [29.5–44.5]). In the early-stage gastric cancer (Fig. 3B), with the same cutoff value and specificity, the AUC of diagnostic efficacy was 0.773 (95% CI [0.701–0.845]) with the sensitivity of 36.1% (95% CI [25.4–48.3]). As shown in Fig. 3C, using the same cutoff value of 1.515 ng/ml, in the independent validation cohort, IGFBP7 could identify gastric cancers with an AUC value of 0.758 (95% CI [0.664–0.852]), a sensitivity of 34.5% (95% CI [22.6–48.7]), and a specificity of 85.5% (95% CI [72.8–93.1]). Similarly, for early-stage gastric cancers diagnosis in the independent

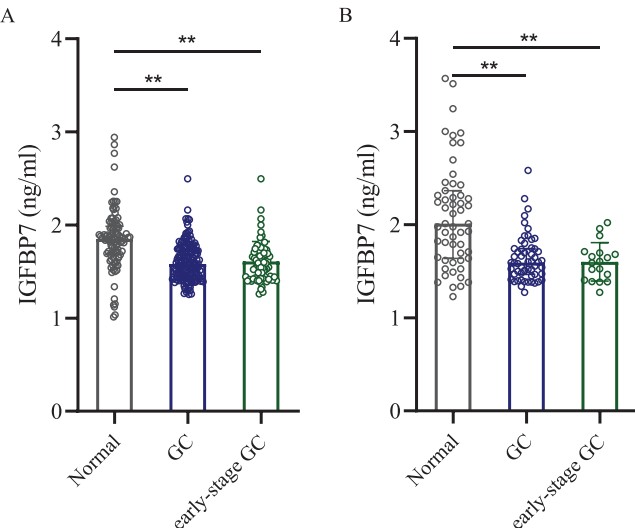

**Figure 2 Dot plots of serum IGFBP7 levels in the normal control group, gastric cancer group and the early-stage gastric cancer group.** (A) Training cohort. (B) Independent validation cohort. IGFBP7: insulin-like growth factor binding protein 7; GC: gastric cancer. Two asterisks (**) mean $p < 0.0001$.

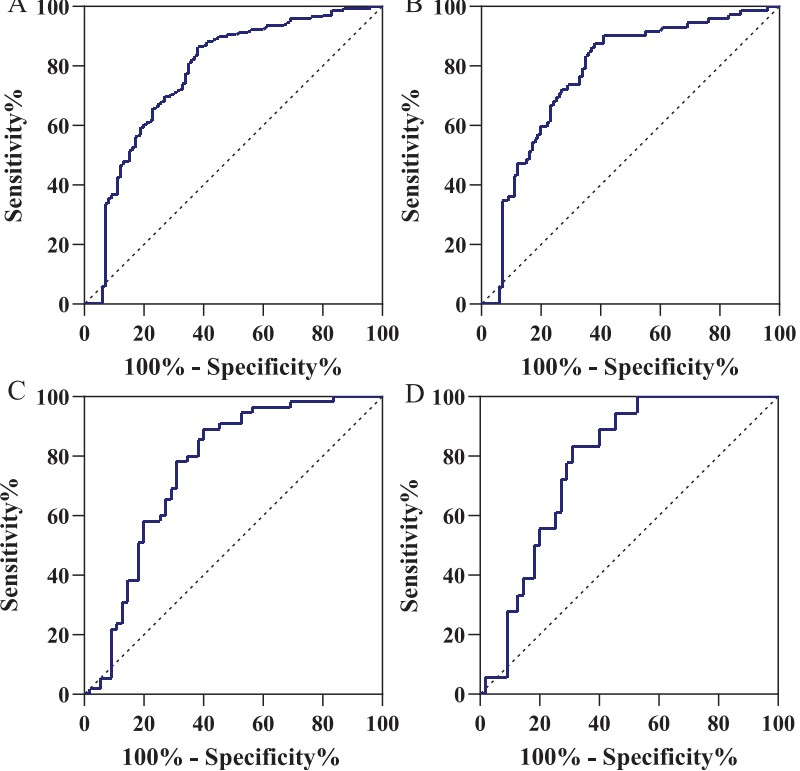

**Figure 3 Receiver operating characteristic curve in the diagnosis of gastric cancer.** (A) Gastric cancer patients *vs* normal volunteers in the training cohort; (B) gastric cancer patients *vs* normal volunteers in the independent validation cohort; (C) early-stage gastric cancer patients *vs* normal volunteers in the training cohort; (D) early-stage gastric cancer patients *vs* normal volunteers in the independent validation cohort. Using the ROC curve, we defined the positive results as an IGFBP7 serum level of less than 1.515 ng/ml.

**Table 2 Result for measurement of IGFBP7 in the diagnosis of gastric cancer.**

| | AUC | SEN | SPE | PPV | NPV | PLR | NLR |
|---|---|---|---|---|---|---|---|
| **Training cohort** | | | | | | | |
| GC vs NC | 0.774 (0.713–0.836) | 36.7% (29.5–44.5%) | 90.0% (82.0–94.8%) | 86.1% (75.5–92.8%) | 45.7% (38.6–52.9%) | 3.67 (1.97–6.82) | 0.70 (0.63–0.79) |
| Early-stage GC vs NC | 0.773 (0.701–0.845) | 36.1% (25.4–48.3%) | 90.0% (82.0–94.8%) | 72.2% (54.6–85.2%) | 66.2% (57.5–73.9%) | 3.61 (1.86–7.01) | 0.71 (0.60–0.85) |
| **Independent validation cohort** | | | | | | | |
| GC vs NC | 0.758 (0.664–0.852) | 34.5% (22.6–48.7%) | 85.5% (72.8–93.1%) | 70.4% (49.7–85.5%) | 56.6% (45.3–67.3%) | 2.38 (1.14–4.96) | 0.77 (0.63–0.93) |
| Early-stage GC vs NC | 0.778 (0.673–0.882) | 33.3% (14.4–58.8%) | 85.5% (72.8–93.1%) | 42.9% (18.8–70.4%) | 79.6% (66.8–88.6%) | 2.29 (0.92–5.72) | 0.78 (0.56–1.09) |

**Note:**
IGFBP7, insulin-like growth factor binding protein 7; GC, gastric cancer; NC, normal controls; AUC, area under the ROC curve; SEN, sensitivity; SPE, specificity; PPV, positive predictive value; NPV, negative predictive value; PLR, positive likelihood ratio; NLR, negative likelihood ratio; the numbers in the brackets mean the 95% confidence interval.

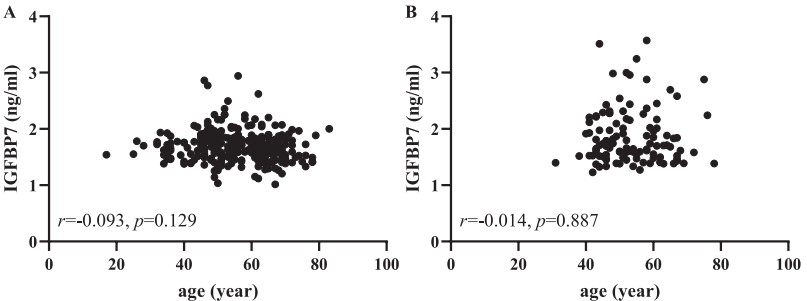

**Figure 4 Pearson correlation analysis between serum IGFBP7 and age in the training (A) and independent validation (B) cohorts.**

validation cohort, the AUC value was 0.778 (95% CI [0.673–0.882]) and the sensitivity was 33.3% (95% CI [14.4–58.8]) with the same specificity (Fig. 3D). Table 2 presented the diagnostic efficacy of IGFBP7 based on positive predictive value (PPV), negative predictive value (NPV), positive likelihood ratio (PLR), and negative likelihood ratio (NLR).

## Correlation between IGFBP7 and clinical characteristics

From Table 1, we found that the control group was younger than the gastric cancer group, especially in the independent validation cohorts. Then we used Pearson correlation analysis to further evaluate the correlation between serum IGFBP7 and age. As shown in Fig. 4, there was no lineal correlation between serum IGFBP7 and age (both $p > 0.05$). As shown in the Fig. S5, there were no correlation between IGFBP7 and any clinical characteristics, including depth of invasion, lymph node metastasis status, distant metastasis and Lauren type in both training and independent validation cohorts (all $p > 0.05$).

As for gender, as shown in the Fig. S6, regardless in male or female, the serum IGFBP7 levels of gastric cancer patients were lower than those of normal volunteers (all $p < 0.05$). In the training cohort, serum IGFBP7 levels of male gastric cancer patients were lower than

those of female patients ($p < 0.0001$). There were no difference of IGFBP7 levels between male and female normal volunteers.

### IGFBP7 in pan-cancers

In order to further explore the expression of IGFBP7 in other cancers, we conducted a pan-cancers analysis using the TCGA and GTEx database. As shown in Fig. S7, IGFBP7 mRNA levels were found to be upregulated in lymphoid neoplasm diffuse large B-cell lymphoma, esophageal carcinoma, glioblastoma multiforme, head and neck squamous cell carcinoma, acute myeloid leukemia, brain lower grade glioma, pancreatic adenocarcinoma, stomach adenocarcinoma, and thymoma. Downregulations were observed mainly in tumors of the urogenital system, lung cancer, and thyroid carcinoma.

### DISCUSSION

Gastric cancer is one of the malignancy diseases with poor survival (*Rawla & Barsouk, 2019*). Early detection, especially the exploration of serum markers, is seen as the best way to prolong cancer-related survival and reduce the cancer burden (*Crosby et al., 2022*). The traditional serum tumor markers such as carcinoembryonic antigen, carbohydrate antigen 19-9 and carbohydrate antigen 72-4 are insufficient for detecting gastric cancer because of the poor sensitivity and specificity (*Katai et al., 2018*). Recently, some novel serum markers has been reported for early detection of gastric cancers. For example, *Roy et al. (2022)* developed an eight-circular RNAs-based panel to assist gastric cancer diagnosis, with the high AUC of 0.82–0.87 in distinguishing early stage gastric cancer patients and healthy controls. *Sun et al. (2021b)* identified inter-alpha-trypsin inhibitor heavy chain 4 as another novel marker for early gastric cancers accompanied by the diagnostic AUC of 0.839 with high sensitivity of 73.08% and high specificity of 94.44%. In our previous study, we found that IGFBP7 might act as a novel serological marker for early detection of esophageal squamous cell carcinoma and esophagogastric junction adenocarcinoma, with the AUC of 0.725 (95% CI [0.633–0.817]) and 0.749 (95% CI [0.644–0.854]), respectively (*Huang et al., 2019*; *Liu et al., 2020*). Extended to the present study, we acknowledged that IGFBP7 might also have early diagnostic value in gastric cancers with an AUC of 0.773 (95% CI [0.701–0.845]) in the training cohort and 0.778 (95% CI [0.673–0.882]) in the independent validation cohort. This finding indicates that IGFBP7 might be another potential serum marker for early diagnosis of gastric cancers.

Insulin-like growth factor (IGF) regulating system contain IGFs, IGF receptors, IGFBPs and IGFBP-rPs. The most important biological and pathological functions of IGFs and IGFBPs include enhancing cell proliferation, inhibiting cell apoptosis and affecting cell transformation in the process of carcinogenesis (*Kasprzak et al., 2017*). IGFBP7, also named as IGFBP-rP1, binds to IGF-1 and IGF-2 with low affinity. In tumor related research, IGFBP7 was first described mainly as a tumor growth suppressing factor (*Oh et al., 1996*). For example, mantle cell lymphoma patients with high IGFBP7 expression would embrace a favorable survival (*Carreras, Nakamura & Hamoudi, 2022*). What's more, *Dang et al. (2021)* revealed that the downregulation of p53/long noncoding RNA IGFBP7 antisense RNA 1/IGFBP7 axis would accelerate the tumorigenesis and progression

of Epstein-Barr virus related B-cell lymphoma. In our present study, serum IGFBP7 protein was also found to be downregulated in gastric cancer patients. The similar result has been published in non-small cell lung cancer patients, which further indicated that the downregulation of IGFBP7 protein was not specific in gastric cancer (*Wang et al., 2013*). However, there are other researches highlighting the tumorigenesis function of IGFBP7. For instance, *Sun et al. (2021a)* found that *via* its receptor CD93, IGFBP7 could promote the angiogenesis of endothelial cells, which might further lead to the tumorigenesis and progression of pancreatic cancer. Other study has shown that increased IGFBP7 also may remodel the tumor microenvironment and promote the progression of esophageal squamous cell carcinoma by activating the transforming growth factor-β1/SMAD signaling pathway (*Li et al., 2022*). In gastric cancer, *Li et al. (2021)* identified that the upregulation of IGFBP7 mRNA may promote the abnormal high expression of lncRNA ubiquitin conjugating enzyme E2 C pseudogene 3, and further promote the progression and metastasis of gastric cancer. High expressions of serum IGFBP7 protein were also found in high-grade soft tissue sarcoma, colorectal cancer, esophageal squamous cell carcinoma, and esophagogastric junction adenocarcinoma (*Benassi et al., 2015*; *Huang et al., 2019*; *Liu et al., 2020*; *Qiu et al., 2020*).

From the TCGA database, we found that IGFBP7 mRNA was overexpressed in gastric cancer tissues in comparison to adjacent normal tissues. However, in the published study, *Liu et al. (2014)* has exhibited that the levels of IGFBP7 protein was lower in 247 gastric cancer patients than the adjacent non-tumor tissues. In our study, we found that serum IGFBP7 protein was down-expressed in gastric cancer patients in comparison to normal volunteers. The potential insistent expression between tissue IGFBP7 mRNA and serum IGFBP7 protein might be due to the degradation of transcription products, translation, post-translation processing and modification.

Recently, a study revealed that the reduced expression of IGFBP7 in gastric cancer cells and tissues might be mainly caused by the aberrant high methylation of IGFBP7 (*Kim et al., 2018*). They also found that the positive rates of IGFBP7 protein was lower in tumor tissues than in the normal tissues, and IGFBP7 protein expression was associated with methylation status but not mRNA expression. From the results of *Gu et al. (2013)*, serum IGFBP7 protein was slightly lower in type 2 diabetes while IGFBP7 DNA methylation levels were increased in these patients. The potential relationship between serum IGFBP7 protein and IGFBP7 methylation might be used to speculate that the downregulation of serum IGFBP7 in gastric cancer might be influenced by the upregulation of IGFBP7 DNA methylation. Yet, this speculation should be further verified in the future.

## CONCLUSIONS

In summary, our present study offers valuable information regarding the diagnostic efficacy of serum IGFBP7 in gastric cancer patients and advocates that IGFBP7 might be a potential serological marker for the detection of gastric cancer. Although our study applied independent validation (*i.e.*, patients enrolled from different time period) to confirm the diagnostic value of serum IGFBP7 in gastric cancer, the sample size was small and bias

should not be ignored. Thus, a larger scale and multicenter study as an external validation should be launched to further validate our results.

### Funding

This work was supported by the Natural Science Foundation of China (81972801); the Science and Technology Special Fund of Guangdong Province of China (STKJ202209069); the Guangdong Basic and Applied Basic Research Foundation-the Enterprise Joint Research Project (2022A1515220116, 2022A1515220180 and 2022A1515220182); the Youth Research Fund Project of Cancer Hospital of Shantou University Medical College (2023A005); the Innovative Team Grant of Guangdong Department of Education (2021KCXTD005); the 2020 Li Ka Shing Foundation Cross-Disciplinary Research Project Fund (2020LKSFG01B and 2020LKSFG01D) and the Science and Technology Planning Project of Shantou City (190413105262902). The funders had no role in study design, data collection and analysis, decision to publish, or preparation of the manuscript.

### Grant Disclosures

The following grant information was disclosed by the authors:
Natural Science Foundation of China: 81972801.
Science and Technology Special Fund of Guangdong Province of China: STKJ202209069.
Guangdong Basic and Applied Basic Research Foundation-the Enterprise Joint Research Project: 2022A1515220116, 2022A1515220180 and 2022A1515220182.
Youth Research Fund Project of Cancer Hospital of Shantou University Medical College: 2023A005.
Innovative Team Grant of Guangdong Department of Education: 2021KCXTD005.
2020 Li Ka Shing Foundation Cross-Disciplinary Research Project Fund: 2020LKSFG01B and 2020LKSFG01D.
Science and Technology Planning Project of Shantou City: 190413105262902.

### Competing Interests

The authors declare that they have no competing interests.

### Author Contributions

- Can-Tong Liu conceived and designed the experiments, analyzed the data, authored or reviewed drafts of the article, and approved the final draft.
- Fang-Cai Wu conceived and designed the experiments, prepared figures and/or tables, and approved the final draft.
- Yi-Xuan Zhuang conceived and designed the experiments, prepared figures and/or tables, and approved the final draft.
- Xin-Yi Huang performed the experiments, prepared figures and/or tables, and approved the final draft.
- Xin-Hao Li performed the experiments, prepared figures and/or tables, and approved the final draft.

- Qi-Qi Qu performed the experiments, prepared figures and/or tables, and approved the final draft.
- Yu-Hui Peng performed the experiments, analyzed the data, authored or reviewed drafts of the article, and approved the final draft.
- Yi-Wei Xu conceived and designed the experiments, analyzed the data, authored or reviewed drafts of the article, and approved the final draft.
- Shu-Lin Chen conceived and designed the experiments, analyzed the data, authored or reviewed drafts of the article, and approved the final draft.
- Xu-Chun Huang conceived and designed the experiments, performed the experiments, analyzed the data, prepared figures and/or tables, authored or reviewed drafts of the article, and approved the final draft.

### Human Ethics

The following information was supplied relating to ethical approvals (*i.e.*, approving body and any reference numbers):

This study was approved by the institutional review board of the Cancer Hospital of Shantou University Medical College and Sun Yat-Sen University Cancer Center.

### Data Availability

The raw data is available in the Supplemental File.

### Supplemental Information

Supplemental information for this article can be found online at http://dx.doi.org/10.7717/peerj.15419#supplemental-information.

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
