# Peer review of "The diagnostic value of serum insulin-like growth factor binding protein 7 in gastric cancer"

_PeerJ, doi:10.7717/peerj.15419_

## Round 0.1 · original submission · Minor Revisions

· Academic Editor

Minor Revisions

According to three reviewers' comments, We think your manuscript needs minor revisions before our next-step decision. Please read the review letter carefully and address all issues that the reviewers have posed. Thanks.

Reviewer 1 ·

Basic reporting

Liu et al. showed that serum IGFBP7 is significantly lower in gastric cancer patients compared with normal controls. This study suggests that IGFBP7 might be used as a potential diagnostic marker for gastric cancer. Several questions need to be addressed:

1. The authors showed that IGFBP7 is significantly lower in gastric cancer patients. Is downregulation of IGFBP7 specific in gastric cancer? Have they checked other cancer types for IGFBP7 levels, at least using TCGA database.

2. Have the authors looked female and male patients separately for IGFBP7 level? Also, is IGFBP7 level different in normal females and males?

3. Can the authors explain how they defined training group and validation group? How did they assign patients in these two groups?

Experimental design

no comment

Validity of the findings

no comment

Additional comments

no comment

Reviewer 2 ·

Basic reporting

no comment

Experimental design

1. Please give a clear explanation of the difference and necessary between the training and validation groups.
2. Figure 2 should add the statistics analysis between GC and Early GC.
3. it is better to show the results of the biopsy tissue by immunohistochemistry or in situ.

Validity of the findings

It is better to add another hallmark as CCN5 to make the diagnosis more convincing.

Additional comments

Please pay attention to the unification of punctuation marks.
for example: at line 62 " 90%(Luo & Li 2019) " space should between 90% and (.
Please revise them in the whole article.

Reviewer 3 ·

Basic reporting

Excellent manuscript with proper scientific structure, wrote in proficient English. Meets the highest quality standards. I found minor issues that could be improved:
Ln 35 – “TCGA” – please expand that abbreviation
Ln 61 – “…”10%” – please cite relevant source
Ln 74 – please rephrase that aim of study. In current form it is too complex sentence.
Ln 100- cite this atlas
Ln 102 – “…patients” – please cite that previous study
Ln 135 – “obeyed” is not a natural language. I recommend “followed” instead
Ln 24 – “…survival” – citation is needed
Ln 278 – “… diseases” – citation is needed

Experimental design

Good technical study. I did not find any flaws in experimental design.

Validity of the findings

Presented results seem reliable, statistically sound. Thesis was answered by results and conclusions. Underlying data has been provided. Conclusions related to the study findings. Interesting manuscript! I highly recommend publication after minor text revisions.

---

## Round 0.2 · accepted · Accept

· Academic Editor

Accept

Congrats! The authors have fully addressed all the concerns of our reviewers. I am pleased to inform the authors that the current version will be sent for the next step of processing before publication.

Reviewer 1 ·

Basic reporting

The authors have addressed most of my questions and the manuscript has improved.

Experimental design

no comment

Validity of the findings

no comment

Reviewer 2 ·

Basic reporting

No comment

Experimental design

No comment

Validity of the findings

No comment

Additional comments

No comment